# Interleukin-18 Binding Protein in Immune Regulation and Autoimmune Diseases

**DOI:** 10.3390/biomedicines10071750

**Published:** 2022-07-20

**Authors:** Seung Yong Park, Yasmin Hisham, Hyun Mu Shin, Su Cheong Yeom, Soohyun Kim

**Affiliations:** 1College of Veterinary Medicine, Konkuk University, Seoul 05029, Korea; paseyo@konkuk.ac.kr; 2Laboratory of Cytokine Immunology, Department of Biomedical Science and Technology, Konkuk University, Seoul 05029, Korea; yasmin.91h@gmail.com; 3System Immunology, Wide River Institute of Immunology, Collage of Medicine, Seoul National University, Hongcheon-gun 25159, Korea; hyunmu.shin@snu.ac.kr; 4Graduate School of International Agricultural Technology, Seoul National University, Pyeongchang 25354, Korea; scyeom@snu.ac.kr

**Keywords:** IL-18BP, soluble anti-IL-18, Th1/Th2 balance, autoimmune diseases, clinical trials

## Abstract

Natural soluble antagonist and decoy receptor on the surface of the cell membrane are evolving as crucial immune system regulators as these molecules are capable of recognizing, binding, and neutralizing (so-called inhibitors) their targeted ligands. Eventually, these soluble antagonists and decoy receptors terminate signaling by prohibiting ligands from connecting to their receptors on the surface of cell membrane. Interleukin-18 binding protein (IL-18BP) participates in regulating both Th1 and Th2 cytokines. IL-18BP is a soluble neutralizing protein belonging to the immunoglobulin (Ig) superfamily as it harbors a single Ig domain. The Ig domain is essential for its binding to the IL-18 ligand and holds partial homology to the IL-1 receptor 2 (IL-1R2) known as a decoy receptor of IL-1α and IL-1β. IL-18BP was defined as a unique soluble IL-18BP that is distinct from IL-18Rα and IL-18Rβ chain. IL-18BP is encoded by a separated gene, contains 8 exons, and is located at chr.11 q13.4 within the human genome. In this review, we address the difference in the biological activity of IL-18BP isoforms, in the immunity balancing Th1 and Th2 immune response, its critical role in autoimmune diseases, as well as current clinical trials of recombinant IL-18BP (rIL-18BP) or equivalent.

## 1. Regulation of IL-18BP

Interleukin-18 (IL-18) was discovered in 1995 as a novel interferon-γ (IFNγ) inducing factor (IGIF), and later it was named IL-18 (also known as IL-1F4) due to the amino acid sequence homology with IL-1β as well as sharing the processing enzyme, caspase-1. Moreover, IL-18 induces IFNγ, including other proinflammatory cytokines in the immune response to pathogens or cell damage. These danger inducements contact cell surface through pathogen-associated molecular patterns (PAMPs) or damage-associated molecular patterns (DAMPs), which in turn bind to the Toll-like receptors (TLR). Then, the activated TLRs trigger myeloid differentiation factor 88 (MyD88) and allow it to make a complex with the interleukin-1 receptor-associated kinase (IRAK) as well as TNF receptor-associated factor (TRAF). This signaling pathway results in oligomerization of inflammasome, and consequently, caspase-1 becomes activated. On the other hand, the translocation of nuclear factor kappa-light-chain-enhancer of activated B cells (NF-κB) induces several proinflammatory cytokines, including pro-interleukin IL-18 (pro-IL-18). The activated caspase is proteolytically cleaved and activates pro-IL-18 to be released from the cell as its mature form; IL-18 (Figure 1, left panel). First, secreted IL-18 binds to the IL-18 receptor alpha chain (IL-18Rα), then recruits the IL-18 receptor beta chain (IL-18Rβ) to initiate the signaling pathway. IL-18 signaling cascades elicited the activation of mitogen-activated protein kinases (MAPKs) and NF-κB, which in turn led to IFNγ and numerous inflammatory gene expressions and enabled continuous loop activation, as the binding of IL-18 to its receptor acts as a downstream activation for NF-κB, (Figure 1, right panel) [1,2,3,4,5,6,7,8].

IL-18 binding protein (IL-18BP) is a natural soluble antagonist, and it binds to the IL-18 ligand with a high affinity. This high-affinity binding of IL-18BP sufficiently blocks the interaction of IL-18 with the IL-18Rα ligand-binding chain on the cell surface and eventually inhibits the IL-18 signaling pathway. Therefore, IL-18BP is considered a potent IL-18 inhibitor (Figure 1, on the top). Furthermore, the production of IFNγ induces IL-18BP to be present as a control role under the term of negative feedback (Figure 2, upper left panel), and IL-18BP is found to be expressed in various organs, such as the spleen, small intestine, stomach, colon, placenta, and lung [7,8,9,10,11,12,13]. This is not surprising, as IL-18 was first described as an IFNγ inducing factor (IGIF) [14], and increasing evidence showed that IL-18 induces IFNγ under various conditions [2,15,16,17]. Notably, IL-18BP was found extracellularly and in 20-folds higher than IL-18 level in healthy sera, which conveys its important regulatory maintenance function [7,18,19,20]. Given that IL-18BP is constitutively expressed from various organs/tissues and secreted into the blood in which it is not anchored on the cell membrane. IL-18BP lacks a transmembrane domain; therefore, it acts as a local as well as systemic IL-18 feedback regulation. As a result, it reduces IFNγ production, avoiding the harmful effect of prolonged inflammatory reactions (Figure 2, upper left panel).

Both IL-18 and IL-18BP occupied a location within chromosome 11 in human genomes. However, each has a distinct cytoband location, q23.1 and q13, respectively [21]. Additionally, the IL-18BP promoter was found to hold several response elements that are required for basal activity. Among them, one site of IFN regulatory factor 1 response element (IRF-E), one site of signal transducer and activator of transcription 1 (STAT1), and two sites of CCAAT-enhancer binding protein beta (CEBP-β), all play a significant role in direct IL-18BP activation in several cell types [12,22]. Based on the existence of these transcription factors, some reports discovered an unexpected expression for IL-18BP irrespective of the presence of IL-18, such as its association with IL-27 [23]. It was found that IL-27 induces the mRNA transcription level as well as the secretion of IL-18BP protein level in a dose-dependent manner. This was accomplished through IL-27-mediated STAT1 activation, which in turn binds to the Gamma-activated sequence (GAS) element within the promoter of IL-18BP, and thus regulates its expression.

Moreover, this transcriptional activation of the IL-18BP promoter regulates the IL-18BP expression by IFNγ and other different stimuli (Figure 2, right panel) [12,24,25]. Notably, the IL-18BP induction was enhanced specifically upon IFNγ, while the levels of the induction varied between monocytes and epithelial cells. These IFNγ induced IL-18BP variation levels were controlled epigenetically through CpG methylation on the promoter region. Therefore, they were diminished in monocytes as the promoter CpG was found to be methylated in contrast to unmethylated in epithelial cells (Figure 2, lower left panel) [26]. As a result, not only CpG methylation, but also other epigenetic modifications can draw the specificity of IL-18BP related to cell types, which is still waiting to be investigated.

## 2. Difference in the Biological Activity of IL-18BP Isoforms

Initially, IL-18BP was discovered in 1999 by the IL-18 ligand affinity chromatographic analysis while looking for a soluble form of IL-18 receptor-ligand binding chain using concentrated human urine as a source of body fluid proteins [7,18,19,20]. It was defined as a unique soluble protein that is distinct from IL-18Rα and IL-18Rβ chain. IL-18BP is encoded by a separated gene and contains 8 exons. The protein encoded by the IL-18BP gene translated into four isoforms in humans, which were produced through the alternative splicing process. These isoforms are IL-18BPa, IL-18BPb, IL-18BPc, and IL-18BPd and have a length of 194, 199, 115, and 163 amino acids, respectively as shown in Figure 3. As previously mentioned, IL-18BP is a soluble neutralizing protein belonging to the immunoglobulin (Ig) superfamily as it harbors a single Ig domain. The Ig domain is essential for its binding to the IL-18 ligand and holds partial homology to the IL-1 receptor 2 (IL-1R2), which is known as a decoy receptor of IL-1α and IL-1β. Among them, two isoforms were found to be active, IL-18BPa and IL-18BPc, since both hold the complete Ig domain. However, their binding affinities to the IL-18 are distinct. IL-18BPa showed an 18 times higher affinity to the IL-18 than IL-18BPc, this is predominantly due to the C-terminal difference. While, on the other hand, IL-18BPb and IL-18BPd lack the essential Ig domain, and thus show no activity in sequestering IL-18. As a result, no inhibition of the IL-18 signal pathway occur [4,5,6,7]. Furthermore, the sequence differences between isoforms point toward different post translational modifications (PTM). Although IL-18BP was reported as a heavily glycosylated protein, isoform b (O95998-3) showed missing conserved amino acids for PTMs. While the two closer isoforms, IL-18BPa and IL-18BPc (O95998, and G3V1C5), hold amino acids for all available PTMs, which are three sites for N-linked glycosylation, site for O-linked glycosylation, and the cysteines for two disulfide bridges, as shown in Figure 3. These modifications may play a key role in the activity of the respective protein/isoform, which also might acknowledge the less or no activity for isoforms other than IL-18BPa. In addition, the amino acid sequence of IL-37 was reported to have some similarity with IL-18 within the binding motif. Therefore, it was suggested that IL-37 might bind to IL-18 receptors and IL-18BP. Yet, these binding and their consequent effect still require additional investigation [27].

## 3. IL-18BP in Immunity Balancing between Th1 and Th2 Immune Response

The anti-inflammatory function of IL-18BP is linked to IL-18 inflammatory activity. Therefore, dysregulation in the balance between IL-18 and IL-18BP results in various disease conditions, which are mainly associated with immunity, an immune response against non-self-antibodies (pathogens), such as viruses, an immune reaction against self-antibodies (autoimmune reactions), and immunity toward cancers [28,29,30,31,32,33,34,35,36,37,38]. As the balance of IL-18 and IL-18BP is disrupted, the immune cell balances are also disrupted, for example, the balance of Th1 and Th2 as well as the balance of Th17 and Treg [28,29,39,40,41,42,43,44]. The balance between Th1 and Th2 immunity at homeostasis or change in this equilibrium due to an immune response to cope with a specific condition is highly regulated by the cellular environment, including levels and types of Th1 and Th2 cytokines [40,41,45]. Negative feedback is required to stop or avoid an overactivated immune response that may increase damage to adjacent healthy cells. IL-18BP is one of the most interesting contributors for the regulation of Th1 and Th2 immune responses (Figure 4) [2,7,28,43,46].

Moreover, as IL-18BP maintains its immune regulation through sequestering IL-18, the IL-18 subsequent activation will be inhibited under various conditions. IL-18 together with IL-12 is associated with immunity against foreign unknown antigens, such as harmful pathogens or pathogenesis of autoimmune diseases. The activated M1 macrophage stimulated Th1 T cells, which in turn induced NK cell activation, cytotoxic T cells, and IFNγ production [47,48,49,50,51,52]. IL-18BP substantially downregulates the production of Th1 cytokines, and thus regulates the balance between Th1 and Th2 responses [7,10].

Additionally, in the absence of IL-12, IL-18 can stimulate Th2 or Th17 response, depending on a costimulatory cytokine which combines with IL-18. For example, when IL-18 is combined with IL-2, they induce mucosal mast cytosis through elevating IL-3 and IL-9 cytokines [53]. Nonetheless, Th2 cytokines are found within allergic conditions and are characterized by increasing IgE production. IL-4 is the major Th2 cytokine, and when combined with IL-18, both induce M2 macrophage and Th2 cells activation resulting in an increase in the production of IgE and Th2 cytokines, for instance, IL-13 and IL-4 [28,54,55,56,57,58,59]. Moreover, IL-18 is suggested to induce an allergic inflammatory response even without IgE help, as is the case in basophils and mast cells activation upon IL-18 with IL-3 [3,57]. Therefore, IL-18BP has also a regulatory effect on Th2 immune response by inhibiting IL-18 activation cascades.

## 4. IL-18BP in Immunity Balancing between Th17 and Treg

Inflammasome-derived IL-18 promotes the production of Th17 cells, cells that induce inflammation, and thus reduce Treg cells, as well as cells that have an immunosuppressor function [60,61,62]. By contrast, when IL-18BP blocks IL-18, subsequently, the Th17 differentiation and Treg will eventually increase and disrupt the balance between Th17 and Treg. The inhibition of Th17 and elevation of Treg cells are beneficial when the superior immune reaction is unwanted, such as in the bone loss process. Additionally, in this case, IL-18BP has been suggested for use in treating osteoporosis [44,63]. Although viruses and cancer cells have granted the benefit of this inhibitory role of IL-18BP to increase Treg cells, and thus evade immune system recognition.

On the one hand, a small number of viruses, such as Poxviruses (C12L, D5L, EVM013, CPXV024), Yatapoxviruses (14L), and Molluscipoxvirus (MC053L, MC054L), encode at least one copy of IL-18BP homologs (named viral IL-18BP; vIL-18BP) [64,65,66,67,68,69,70]. The vIL-18BP was found to have similar human IL-18 binding sites [65], explaining the viral dissemination and modulation of the host antiviral response, which result in immune evasion. In addition to its ability to bind and neutralize IL-18, it was found to reduce IFNγ production in vitro studies [64,65,66,69,70,71]. Moreover, deleting vIL-18BP in these viruses potentially improves the respective live attenuated vaccines [72].

Moreover, mycobacterium tuberculosis (MTB), which caused tuberculosis (TB) pulmonary illness, has been known for its immune evasion, and thus still has serious clinical concerns. Lately, IL-18, as well as IL-18BP, has been found to be associated with active TB infection [73]. In this regard, IL-18 was identified as one of the pivotal cytokines that regulate macrophage cells during this infection in supplement with IL-12, IL-27, IFNγ, and TNFα [74]. Recently, serum levels of IL-18BP were shown as one of the potential biomarkers to discriminate between latent and active TB, further between active TB and non-mycobacterial community-acquired pneumonia (CAP) [73,75,76].

On the other hand, IL-18 is highly expressed in several cancers, within its microenvironment IL-18BP acts as a barrier for the cancer niche not only by increasing Treg, but also by attenuating natural killer (NK)-mediated killing [77,78,79]. In addition to the upregulation of IL-18BP in inflammatory conditions as mentioned earlier, IL-18BP is upregulated in various cancer types, such as breast [80], pancreas [33], prostate [31], and ovarian [34]. Therefore, IL-18BP has been suggested as an immunotherapy potential target for many cancers [77,81]. Initially, a clinical trial started with a recombinant IL-18 (rIL-18: SB-485232). However, during phase II, this recombinant cytokine demonstrated inadequate anti-tumor activity, especially with metastatic cancers. Therefore, it was further tested in combination with other therapies (ClinicalTrials.gov Identifier: NCT00107718) [82,83]. This unexpected lower efficiency is mostly due to the production of IL-18BP in the cancer microenvironment that neutralizes exogenous IL-18, as well. Recently, Zhou et al. have engineered an effective anti-tumor decoy-resistant IL-18 (DR-18). Namely, in contrast to the wild type rIL-18, it is characterized by the reinstate NK cell function, and thus promotes cancer clearance, suggesting the enhancement of anti-tumor immunity with other immunotherapies [78,79,84]. Still, further studies are needed to evaluate the newly developed DR-18 agonist for several cancer types. This finding encourages researchers and pharmaceutical companies to unveil more similar molecules that can effectively facilitate cancer clearance.

Nevertheless, it was shown that the absence of IL-18BP resulted in excessive IL-18 activity, which leads to strong and uncontrolled immune reactions, such as in the case of acute renal injury, cholestatic liver injury, and acute live injury [85,86,87,88]. In line with this fact, the loss of IL-18BP function seems to be fatal for humans, considering that in 2019 a case report confirmed the death of a child with a loss-of-function mutation in IL-18BP (homozygous 40-nucleotide deletion) upon hepatitis A virus infection [89]. As in this recently mentioned case, the infection was followed by an endless NK activation due to continuously released IL-18 without its sequestering by the loss of IL-18BP function causing cytotoxicity for healthy hepatic cells, thus disrupting immunity, and establishing toxicity and lethality. Therefore, the absence of IL-18BP reveals that the baseline levels of IL-18 can be destructive to humans, and thus it should be examined wisely and monitored continually.

## 5. IL-18BP in Autoimmune Diseases

IL-18/IL-18BP imbalance is highly linked to immunologically mediated diseases, especially diseases that have a pathological role of IFNγ, such as macrophage activated syndrome (MAS) [2,32,90,91,92,93,94]. MAS is a life-threatening condition that is not mainly a syndrome on its own, but is also found to be associated with other infectious and autoinflammatory diseases [2,32,93,95,96,97,98,99,100]. Various infections developed MAS, including Epstein–Barr virus, cytomegalovirus, and herpes virus, and infections with other pathogens, such as intracellular bacteria and parasites, as well as numerous lymphomas [32,90,91,92,93]. Moreover, MAS could be present as a form of secondary hemophagocytic lymphohistiocytosis (HLH), in addition to the rare and serious rheumatic diseases, such as adult-onset Still’s disease (AOSD) and its counterpart in children and systemic juvenile idiopathic arthritis (sJIA) [94,101,102,103,104,105]. In both conditions, secondary HLH and AOSD/sJIA, a significant elevation of IL-18 was validated and found to be correlated to disease activity and severity [94,104,106,107,108,109,110,111,112,113,114,115,116,117,118,119,120,121,122,123,124,125]. MAS and IL-18 serum levels both represent the shared pathogenic features between these two different disease conditions. Under active MAS, IL-18 induces T cells expansion, and thus enhances the production of IFNγ and antigen-presenting cells (APCs) activation, predominantly dendritic cells (DCs), which in turn support the activation of macrophages. Consequently, this activation results in prolonged activation of these cells along with their inflammatory mediators, such as cytokine storm [106,107,108]. Therefore, increasing evidence showed that MAS-associated AOSD/sJIA had higher levels of IL-18 when compared to patients with no MAS association, IL-18BP or equivalent, which are suggested as a potential therapeutic option to neutralize IL-18 for MAS-associated conditions [95,120,126,127,128]. This suggestion was translated into drugs that are currently under distinct phases of clinical trials, which were summarized in Table 1.

Tadekinig-α is a human recombinant IL-18BP (rhIL-18BP) with an IL-18 high affinity and has been investigated in two serious inflammatory conditions that are associated with unusual elevation of IL-18 plasma levels; AOSD/sJIA and a refractory HLH bearing a gain of function mutation named NLR family CARD domain containing 4 (NLRC4)-related MAS. To date, promising results showed a favorable response toward Tadekinig-α in AOSD/sJIA and NLRC4-related MAS, which are currently under phase II and III, respectively (ClinicalTrials.gov Identifier: NCT02398435, NCT03113760, NCT03512314) [104,129,130,131]. Notably, in phase II clinical trial on Tadekinig-α, 50% of the AOSD cohort reached normal body temperature and reduced C-reactive protein (CRP) levels within 3 weeks under treatment, confirming the neutralization effect of the free bioactive IL-18 [129]. More recently, a case report revealed undetectable serum levels of the free IL-18 after only 2 h upon first administration of Tadekinig-α, while again increased when the administration of Tadekinig-α was discontinued [130].

MicroRNAs (miRNAs) are known as short non-coding sequences of RNA, which show a repression function. Notably, the reported miRNA (miRNA-134) that targeted IL-18BP, was found to be elevated in AOSD plasma and correlated with the activity of the disease [132]. It is well known that changes in miRNAs are considered as post-transcription gene regulators, and participate in human disorder pathogenesis [133]. Therefore, miRNA-134 can be an AOSD biomarker as well as a target for augmentation therapy. On the other hand, HLH patients carrying the NLRC4 mutation were examined under the Tadekinig-α regimen earlier than AOSD/sJIA, demonstrating a successful treatment in HLH holding NLRC4 genetic background [131,134,135].

Although IL-18/IL-18BP has significantly dysregulated in the abovementioned MAS-associated diseases, the imbalance of IL-18/IL-18BP has also been noticed in other inflammatory diseases, including rheumatoid arthritis (RA), psoriasis, asthma, lupus erythematosus, multiple sclerosis, atherosclerosis, renal and liver injury, inflammatory bowel disease (IBD), Crohn’s disease (CD), organ transplant rejection together with Graft versus host disease (GvHD), and most recently in pyogenic sterile arthritis, pyoderma gangrenosum, and acne (PAPA) syndrome [15,37,86,136,137,138,139,140,141,142,143,144,145,146,147,148], where cytokine imbalance is a major feature of these autoimmune disorders. The refractory monogenic inflammatory disorder, PAPA, resulted from a dominant mutation with the proline-serine-threonine phosphatase interacting protein 1 (PSTPIP1) gene and is characterized by acne and skin ulceration. In addition, the autoimmune neutrophilic destruction of joints and skin is the major clinical presentation of PAPA, but it was not associated with MAS [149,150]. Moreover, a recently published study found that IL-18 is elevated in the patient serum and this elevation was highly associated with the disease outcome, again with no MAS risk [148].

Therefore, IL-18 elevation and depletion in IL-18BP function is a biomarker for most of these diseases. However, targeting IL-18 by IL-18BP or equivalent is yet to be fully explored. Humanized IgG1 monoclonal antibody (anti-IL-18), GSK1070806, is instantly in the development phase for treating a variety of inflammatory conditions, atopic dermatitis (AD) and IBD in phase I, delayed graft function, CD, and Behcet’s disease in phase II (ClinicalTrials.gov Identifier: NCT04975438, NCT01035645, NCT02723786, NCT03681067, NCT03522662) [151]. In addition to Tadekinig-α and GSK1070806, a long-acting antibody drug targeting IL-18 and APB-R3 is developed for inflammatory autoimmune disease treatment in preclinical stages, and will soon enter phase I clinical trial [152].

Improving IL-18 targeting by allowing it to act for a longer time or with superior affinity appears to be promising and beneficial, taking into consideration sequence and structural features that affect the binding interaction. A recent structural study aimed to characterize the IL-18BP sequestration mechanism and reveal a novel disulfide-linked boundary, resulting in tetrameric assembly between IL-18BP and IL-18 [153]. During their study, due to N-terminal heavy glycosylation, they produce hIL-18BPΔN, an hIL-18BP that lacks N-terminal residues 63-194 and is treated with Endoglycosidase H. This form has lower glycosylation and still has the disulfide bridge, which shows a comparable IL-18 sequestering ability with high affinity, but at a higher rate. Therefore, hIL-18BPΔN might be a promising IL-18 neutralizing drug (see Figure 3 and Figure 4).

IL-18 has been involved in the injury of several organs as well as in potentially fatal conditions, which are exemplified by a cytokine storm. Lately, after the emergence of the COVID-19 crisis, heavy-handed cases of SARS CoV-2 infection showed a cytokine storm that derives tissue damage [154,155,156,157]. The COVID-19 mediated cytokine storm showed that elevated IL-18 is one of the cytokines involved in the storm. Moreover, due to its remarkable elevation, IL-18 was recognized as a biomarker for disease severity [154,155,157,158,159]. Therefore, IL-18BP has been suggested as a promising therapy for COVID-19. Yet, there are no pre or clinical trials in this regard [17,160,161]. Furthermore, the lack of proper and well-characterized animal models slows down the movement for further discovering and understanding IL-18BP as a potential therapeutic option for a wide range of disorders.

## Figures and Tables

**Figure 1 biomedicines-10-01750-f001:**
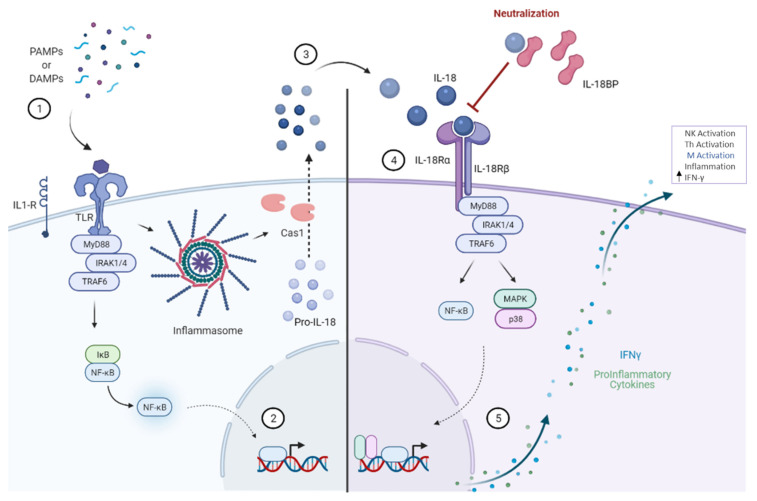
IL-18BP Biological Role. Left panel, ① danger stimuli, such as pathogen-associated molecular patterns (PAMPs) and damage-associated molecular patterns (DAMPs), activate the Toll-like receptors (TLR) signaling pathway that triggers MyD88, TRAF, inflammasome oligomerization that leads to subsequent activation of caspase-1 (Cas1). Consequently, the translocation of the NF-κB takes place and leads to ② the production of proinflammatory cytokines, including inductions of the transcriptional activation of pro-interleukin IL-18 (pro-IL-18), which is proteolytically cleaved and converted to its mature active form via the active caspase-1 (Cas1), then ③ IL-18 is secreted from the cell. Right panel, ④ when secreted IL-18 binds to the cells expressing IL-18R (mainly monocytes, NK cells, and T cells) and activates the respective signaling pathways, for example, MAPKs and NF-κB activation, which in turn leads to the ⑤ expression of proinflammatory molecules and induce the production of IFNγ. Next, IFNγ activates macrophages and allows them to produce inflammatory cytokines. This IFNγ induction and the activation cascade of IL-18 is inhibited by its natural neutralizing soluble inhibitor, interleukin IL-18 binding protein (IL-18BP).

**Figure 2 biomedicines-10-01750-f002:**
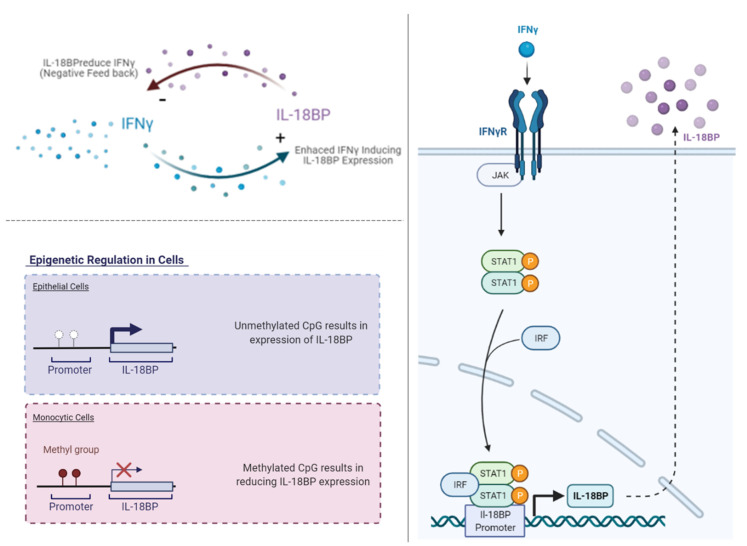
IL-18BP expression induced by IFNγ. IFNγ production is induced due to cytokines that are released in response to infection, tissue damage, activation of pattern recognition receptors (PRRs) or reactive antigen stimulation. Left side (up), the upregulated IL-18BP triggers the negative feedback loop, which in turn allows IL-18BP to inhibit the production of IFNγ. Left side (down), the epigenetic regulation of IL-18BP in various cells, methylation at the IL-18BP promoter region controls its expression upon IFNγ stimulation. Epithelial cells have no CpG methylation; therefore, their IL-18BP inducibility is strong and higher than monocytic cells, which have CpG methylation. Right side, IFNγ signaling starts when it binds to its receptor IFNγR, which consists of two subunits, IFNγR1 and IFNγR2. Later, this binding induces the Janus kinase (JAK) signal transducer and activator of transcription (STAT) signaling, particularly STAT1 makes a dimer, and then cooperates with IFNγ-induced interferon regulatory factors (IRFs), resulting in translocation into the nucleus. IL-18BP promoter holds several response elements, among them IFN regulatory factor 1 response element (IRF-E) and STAT1. These transcriptional activations of the promoter induce IL-18BP expression.

**Figure 3 biomedicines-10-01750-f003:**
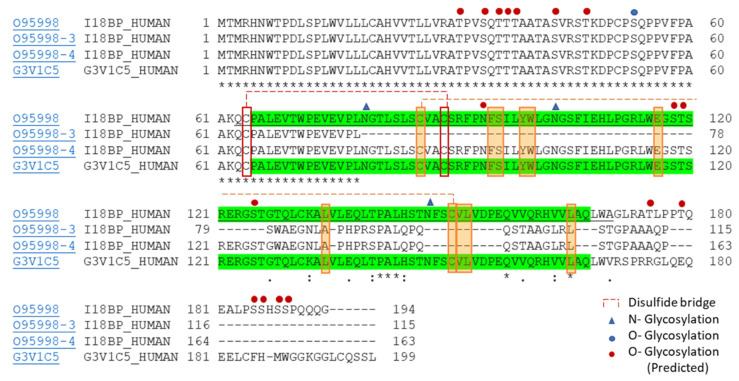
IL-18BP isoforms. Multiple sequence alignment for the isoforms of human IL-18BP. PTM features are indicated as glycosylation sites, which are displayed above the sequence (N-linked; Triangle, O-linked; circle, blue for confirmed, red for predicted O-glycosylation). Disulfide bridges are exposed as dashed lines (red and orange). Residues with homology are indicated with an asterisk (*) below the residue. Residues shearing identity with the virus IL-18BP are highlighted within an orange color shaded box. The domain of Ig-like C2-type is highlighted in green (starts at 65 and ends at 166 with a length of 102aa based on the sequence of IL-18BPa). Uniport accession number; IL-18BPa (O95998 I18BP_HUMAN/NP_766630.2), IL-18BPb (O95998-3 I18BP_HUMAN/NP_001138527.1), IL-18BPd (O95998-4 I18BP_HUMAN/NP_766632.2), and IL-18BPc (G3V1C5 _HUMAN/NP_005690.2). (:) Conservative residue; (.) Semi-conservative residue; ( ) Non-conservative residue.

**Figure 4 biomedicines-10-01750-f004:**
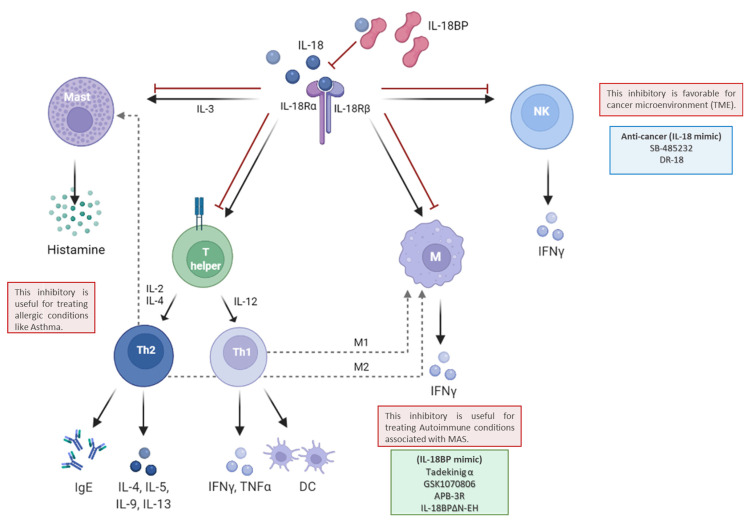
IL-18BP inhibiting IL-18 target cells and therapeutic interventions. IL-18 assists several immune cells, it stimulates NK to produce IFNγ, which activates macrophages and allows them to produce inflammatory cytokines. In addition, it participates in the Th1 and Th2 response, depending on cytokine partners, when IL-18 is combined with IL-12 it facilitates the Th1 immune response, which in turn activates DCs and releases inflammatory cytokines, such as IFNγ and TNFα. In response to Th1 activation, M1 macrophages induce IL-18 and supply the loop. Alternatively, in the absence of IL-12, and when IL-18 is combined with IL-2 and/or IL-4, together they facilitate Th2 immune response, which in turn secreted IgE and numerous Th2 cytokines, such as IL-4, IL-5, IL-9, and IL-13. In response to Th2 activation, M2 macrophages and/or mast cells become activated, and then the production of histamine takes place (allergic response), which is also facilitated when IL-18 is combined with IL-3. When IL-18BP sequestered IL-18, the following activated cascade is inhibited. Therefore, IL-18BP can play potential roles in both Th1 and Th2 autoimmune diseases. Boxes indicated the main pathological condition (RED), potential therapeutic options (BLUE; anticancer molecules in the clinical trial, GREEN; IL-18BP or equivalent; Tadekinig-α, GSK1070806, and APB-3R are in clinical and pre-clinical trials, IL-18BPΔN-EH showed higher affinity than full-length IL-18BP and suggested for use in treating autoimmune diseases).

**Table 1 biomedicines-10-01750-t001:** Clinical trials associated with IL-18.

Condition/Disease	Clinical Phase	Clinical Trial Status	No. of Participants	Type of Intervention or Treatment	NCT Number
Still’s Disease, Adult-Onset	Phase II	Completed	23 participants	Biological: Tadekinig alfa (recombinant human IL-18 binding protein)	NCT02398435
XIAP Deficiency, NLRC4-MAS	Phase III	Recruiting	10 participants	Drug: Tadekinig alfa	NCT03512314
XIAP Deficiency, NLRC4-MAS	Phase III	Recruiting	10 participants	Drug: Tadekinig alfa	NCT03113760
Inflammatory Bowel Diseases	Phase I	Completed	78 participants	Drug: GSK1070806 (block IL-18)	NCT01035645
Kidney Transplantation (Status Post)	Phase II	Terminated *	7 participants	Drug: GSK1070806 + others	NCT02723786
Melanoma	Phase II	Completed	64 participants	Drug: SB-485232 (rhIL-18)	NCT00107718
Dermatitis, Atopic	Phase I	Recruiting	48 participants	Drug: GSK1070806	NCT04975438
Crohn Disease	Phase I Phase II	Completed	5 participants	Drug: GSK1070806	NCT03681067
Behcet’s Disease	Phase II	Unknown #	12 participants	Drug: GSK1070806	NCT03522662

* Lack of efficacy. # Study has passed its completion date and status has not been verified in more than 2 years.

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
