# Peer review of "Interleukin-18 Binding Protein in Immune Regulation and Autoimmune Diseases"

_biomedicines, 2022, doi:10.3390/biomedicines10071750_

Round 1

Reviewer 1 Report

The review article by Park et al, provides a comprehensive overview of the role of IL-18 in immune regulation. The various pathways in which IL-18 plays a crucial role are well described. The figures are very clear and literature is well cited. However, the manuscript requires major English editing and grammatical checks throughout, including the figure captions. 

Author Response

We appreciate your recommendation, we are proceeding with English editing and proofreading for the entire manuscript.

Reviewer 2 Report

The manuscript is well structured and developed. However, the authors do not fully and accurately address the clinical trials of Interleukin-18 binding protein. The authors only point out the clinical trials, without a complete description that includes the clinical phase, the condition or disease addressed, the status of the clinical trial, the number of participants, etc.

Therefore, authors should approach clinical trials through a table that includes the following information: clinical phase, condition/disease, clinical trial status, number of participants, study arms, and type of intervention or treatment. This information will enable the reader to establish the current status of IL-8 binding protein therapy.

Author Response

We appreciate your recommendation, kindly note that Listing and comparing clinical trials and describing each comprehensively is not our scope. Moreover, we mentioned the relevant clinical trials and provided their IDs so that readers who want to know more can check the original data. However, we provided a table of clinical trials with brief information including clinical phase, condition/disease, clinical trial status, number of participants, type of intervention or treatment, and NTC number.